# Sodium Pre-Intercalation-Based Na_3_-δ-MnO_2_@CC for High-Performance Aqueous Asymmetric Supercapacitor: Joint Experimental and DFT Study

**DOI:** 10.3390/nano12162856

**Published:** 2022-08-18

**Authors:** Anis Ur Rahman, Nighat Zarshad, Wu Jianghua, Muslim Shah, Sana Ullah, Guigen Li, Muhammad Tariq, Asad Ali

**Affiliations:** 1Institute of Chemistry and BioMedical Sciences, School of Chemistry and Chemical Engineering, Nanjing University, Nanjing 210023, China; 2Department of Polymer Science, School of Chemistry and Chemical Engineering, Southeast University, Nanjing 211189, China; 3National Laboratory of Solid State Microstructures, Collaborative Innovation Center of Advanced Microstructures, College of Engineering and Applied Sciences, Nanjing University, Nanjing 210093, China; 4Department of Chemistry, Faculty of Chemical and Life Sciences, Abdul Wali Khan University, Mardan 23200, Pakistan; 5Department of Chemistry and Biochemistry, Texas Tech University, Lubbock, TX 79409, USA; 6Department of PCB, Bayazid Rokhan Institute of Higher Studies, Kabul 1002, Afghanistan

**Keywords:** manganese oxide, pre-intercalation, specific capacitance, wide potential window, asymmetric aqueous supercapacitor, DFT calculation

## Abstract

Electrochemical energy storage devices are ubiquitous for personal electronics, electric vehicles, smart grids, and future clean energy demand. SCs are EES devices with excellent power density and superior cycling ability. Herein, we focused on the fabrication and DFT calculations of Na_3_-δ-MnO_2_ nanocomposite, which has layered MnO_2_ redox-active sites, supported on carbon cloth. MnO_2_ has two-dimensional diffusion channels and is not labile to structural changes during intercalation; therefore, it is considered the best substrate for intercalation. Cation pre-intercalation has proven to be an effective way of increasing inter-layered spacing, optimizing the crystal structure, and improving the relevant electrochemical behavior of asymmetric aqueous supercapacitors. We successfully established Na^+^ pre-intercalated δ-MnO_2_ nanosheets on carbon cloth via one-pot hydrothermal synthesis. As a cathode, our prepared material exhibited an extended potential window of 0–1.4 V with a remarkable specific capacitance of 546 F g^−1^(300 F g^−1^ at 50 A g^−1^). Moreover, when this cathode was accompanied by an N-AC anode in an asymmetric aqueous supercapacitor, it illustrated exceptional performance (64 Wh kg^−1^ at a power density of 1225 W kg^−1^) and incomparable potential window of 2.4 V and 83% capacitance retention over 10,000 cycles with a great Columbic efficiency.

## 1. Introduction

The continuous depletion of non-renewable energy resources as a consequence of rapid economic development around the globe has forced researchers to seek technological evolution for the future to satisfy the demands of efficient and renewable energy. In this regard, the supercapacitor stands out as the most promising candidate owing to its fast charge/discharge rate, supreme cyclic stability [1], as well as high power density [2]. However, the lower energy density of the currently employed supercapacitors is a limiting factor for their practical applicability [3]. Therefore, it is indispensable to boost the energy density of supercapacitors without impacting their stability and power density. This can be accomplished by either enhancing the specific capacitance or cell voltage. The construction of an asymmetric supercapacitor is a viable route for achieving high cell voltage, where the negative electrode is present at a lower potential, while the positive electrode is present at a higher potential and operates in a different potential window [4,5]. This results in improved cell voltage and energy density of the material. However, the maximum cell voltage obtained so far lies between 1.4 and 2.0 V, which is still not enough to practically fulfill the energy demands [6,7]. Therefore, the quest of developing asymmetric supercapacitors with a voltage range higher than 2.0 V is still an untangled issue [8].

As a cathode material, MnO_2_ has been a fascinating choice because of its large theoretical specific capacitance value (~1370Fg^−1^) and wide potential window (~1.0 V) [9,10,11,12]. MnO_2_ can be found in a variety of crystal structures, such as α, β, γ, δ, and λ MnO_2_. The electrochemical properties of α, β, and γ-MnO_2_ are associated with their chain/tunnel type crystal structure facilitating electron transport for achieving high specific capacitance [13]. Moreover, the layered sheet-like structure of δ-MnO_2_ makes it suitable for the intercalation of various cations, so as to increase the interlayer spacing, improving the migration rate of electrons, which increases the specific capacitance [14,15].

Pre-intercalation of MnO_2_ has acquired much attention in recent years as a viable strategy to escalate the electrochemical efficiency of MnO_2_-based supercapacitors [16]. Pre-intercalated MnO_2_ has several ions intercalated into the redox channel and the interlayer of MnO_2_ before electrochemical measurements [17]. Chemical bonding allows these intercalated ions to interact with the host framework and the incorporated carrier ions on an electrostatic and physical level, which has a significant impact on the intrinsic structure of MnO_2_ and the carrier ion transport kinetics [18,19]. High specific surface area and optimized channels available for quick and reversible ion injection and extraction play a key role in achieving excellent specific capacitance, increased energy density, and extended potential window [20].

To elucidate the availing influence of pre-intercalation, some published works are mentioned below. It has been proposed [21] that pre-intercalated K^+^ ions inside 2 × 2 tunnels of MnO_2_ would improve Li^+^ diffusivity by increasing electronic conductivity and interlayer spacing and electrostatic interactions between the inserted Li^+^ ions and host anions. On the other hand, they are thought to have a big impact on the activation barrier for Li^+^ hopping in the layered lithium transition metal oxides [22,23,24]. The charge shielding property of crystal water lowers electrostatic interactions between the carrier ions and host anions, improving carrier ion diffusion kinetics in MnO_2_ cathodes, according to the literature [25,26].

We intended to construct ultra-thin Na^+^ ions pre-intercalated MnO_2_ cathode material on carbon cloth (CC), as shown in Figure 1, a high-performance supercapacitor with an extended working potential window of 0–1.4 V, which has a very high reversible capacitance of 560 F g^−1^. The high capacitance is due to the pre-intercalated Na^+^ ions in the MnO_2_ nanosheets. Furthermore, an asymmetric aqueous supercapacitor with Na_3_-MnO_2_ positive and N-AC negative electrode was constructed, demonstrating a wide potential window of 2.4 V with a high energy density of 64 Wh kg^−1^ at a power density of 1225 W kg^−1^ and strong cycling stability of 83% capacitance retention. Our sample has a substantially greater content of pre-intercalated Na ions, which was prepared using a simple one-pot technique. Furthermore, the quick and effortless procedure, the low hydrothermal temperature, and the shorter time spent formulating our samples make the materials and process strong and significant in comparison to all the MnO_2_-based electrodes.

## 2. Materials and Methods

All the chemical reagents were of analytical grade and used after purchase without any purification. All chemicals were Aladdin reagents and purchased from Shanghai Macklin Biochemical Co., Ltd., Shanghai, China.

The crystalline structures were characterized by Bruker D8 super speed X-ray diffractometer (XRD) with Cu Kα radiation. The morphology of samples was observed with scanning electron microscopy (SEM) by (NAVO NanoSEM450 electron microscope FEI, USA), transmission electron microscopy (TEM) by JEOL 2010 transmission electron microscope, and high-resolution transmission electron microscope (HRTEM). Chemical characterization was performed by X-ray photoelectron spectroscopy (XPS) with an ESCALAB 250Xi spectrometer (Thermo Fisher, Loughborough, UK). Energy-dispersive X-ray spectroscopy (EDS) and HRTEM were used for elemental mapping.

### 2.1. Treatment of Carbon Cloth (CC)

CC was cut into a 4 cm × 2 cm rectangular strip, soaked, and sonicated in deionized water for 10 min. The CC was then sonicated in ethanol for 10 min. Acetone was used to repeat the process. Water, ethanol, and acetone were used to sonicate three times each. Finally, the carbon cloth was dried overnight at 60 °C in an oven.

### 2.2. Synthesis of Na_3_-MnO_2_

KMnO_4_ (0.5 mmol, 0.079 g) and Na_2_SO_4_ (21 mmol, 3 g) were dissolved using 50 mL of deionized water, and the solution was stirred for 1 h at room temperature. The clear solution was poured into a 100 mL PTFE liner enclosed by a stainless-steel autoclave, into which CC was placed afterward. The hydrothermal reaction took place at 120 °C for 1.5 h. The sample grown on the CC was rinsed three times with DI water and, finally, with ethanol. The mass loading of Na_3_-MnO_2_ on the CC was 1.2 mg cm^−2^. Pure MnO_2_ was produced using the same technique but without the addition of Na_2_SO_4_.

## 3. Results

The Na^+^ pre-intercalated MnO_2_ on CC was synthesized by a reasonably simple hydrothermal technique. To ensure structural integrity, the CC chosen as a substrate for growing samples prevented the addition of conductive additives and the polymer binder. CC, with its significant mechanical stability and flexibility, is a strong contender for binder-free electroactive material development [27,28].

Figure 1 shows an X-ray diffraction (XRD) investigation. δ-MnO_2_ (**JCPDS card no.80-1098**) can be given from the inset pattern because of the four main representative peaks detected at 12.4°, 25°, 37°, and 65.5° indexed with (001), (002), (111), and (020) having an interlayer spacing of 0.7 Å [27,29]. The XRD patterns show the genesis of pure δ-MnO_2_, but the low intensity and broadness of the peaks indicate the formation of nano-sized crystallites with poor crystallinity. There is no discernible change in the position of diffraction peaks after the addition of Na ions, demonstrating that pre-intercalated Na^+^ ions do not affect the crystal structure of MnO_2_.

Field-emission scanning electron microscopy was used to observe the morphology and structure of all the Na_3_-MnO_2_ samples. Figure 2 shows that the morphology did not change after the pre-intercalation of Na^+^ to MnO_2_. Furthermore, high-magnification SEM images revealed that the Na_3_-MnO_2_ is made up of ultra-thin uniform nanosheets of about a few nm thicknesses that are uniformly grown on the carbon cloth (Figure 2c). Nanosheets of Na_0_-MnO_2_ (Figure 2c) are thickner than the nanosheets of Na_3_-MnO_2_ (Figure 2f), which is thought to be helpful for ions insertion/extraction. The specific surface areas and electron-ion transfer distance affect the thickness of the nanosheet, which has a pronounced impact on the electrochemical properties of Na_x_-MnO_2_ [30]. The EDS spectrum of Na_3_-MnO_2_ is shown in Appendix A. The presence of the Na element in MnO_2_ nanosheets can be seen in the elemental mapping of the energy-dispersive X-ray spectroscope in Appendix A. EDS analysis demonstrated that only Mn, Na, C, O, and K elements were present in the Na_3_-MnO_2_. This analysis proves that Na^+^ ions successfully pre-intercalated and occupied the interlayer sites of the layered δ-MnO_2_

TEM analysis was used to investigate the structural details in greater depth and to assess the impact of the introduction of Na ions on the morphology of the Na_3_-MnO_2_ electrode. Figure 3 depicts TEM representations of Na_3_-MnO_2_ and MnO_2_ electrodes with and without Na ions. The uniform MnO_2_ nanostructures are composed of ultra-thin nanosheets, as shown in Figure 3a,c. The pre-intercalated Na ions decrease the grain size of MnO_2_ nanosheets to a minuscule. Consequently, the active sites for ion adsorption are increased due to exceptional electrochemical performance. The creation of oxygen vacancies might emphasize the intrusion of electrolytes ions, leading to improved redox reaction and boosted conductivity. Figure 3b is the HRTEM image of Na_3_-MnO_2_ nanosheets verifying the XRD data, as the lattice space in the central region is 0.7 nm, which correlates with the crystal plane (001) of δ-MnO_2_. The high-angle annular dark-field (HAADF) imaging in Figure 3e indicates a sustained arrangement of atomic species in the crystal phase of MnO_2_, which also proves the presence of pre-intercalated Na ions in the layers of δ-MnO_2_.

Spherical aberration-corrected electron microscopy revealed the atomic resolution structure of the Na_3_-MnO_2_ (Figure 3g,h). In Figure 3h, the white spots reflect the position of Mn atoms as seen from the <001> zone axis.

The Na_3_-MnO_2_ was analyzed by X-ray photoelectron spectroscopy (XPS). The survey spectrum shown in Figure 4a indicates the presence of C, O, Mn, and Na elements. Figure 4b reveals the high-resolution spectrum of Mn 2p. The peaks are pinpointed at binding energies of 642.3 eV and 654 eV, which correlate with the binding energies of Mn 2p_3/2_ and Mn 2p_1/2_, respectively, with a difference of 11.7 eV. This stipulates that the oxidation state of Mn in the Na_3_-MnO_2_ is +4, which is consistent with the previous literature [31,32,33,34]. Figure 4c shows high-resolution XPS spectra of Na 1s, which has a binding energy of 1071.4 eV [35,36,37]. It authenticates the successful intercalation of the Na^+^ into the Na_3_-MnO_2_.

To evaluate the electrochemical efficiency of the designed electrodes in a module of a three-electrode system, the electrochemical measurements were carried out on a VMP3 biologic workstation. This system consists of Na_0_-MnO_2_ and Na_3_-MnO_2_ as a working electrode and a Hg/HgO as a reference electrode and is completed with platinum foil as a counter electrode. We recorded the galvanostatic charge/discharge (GCD) measurements, electrochemical impedance spectroscopy (EIS), and cyclic voltammetry (CV) with 1M Na_2_SO_4_ solution as the electrolyte.

Figure 5a displays the characteristics of CV curves of Na_3_-MnO_2_ at the enlarged working potential range of 0–1.4 V at different scan rates (2, 5, 10, 20, and 50 mV s^−1^). All the graphs represent a nearly rectangular shape comprising a noticeable redox peak at about 0.8 V. The current density increases correspondingly with the increase in scan rate, which is typical of MnO_2_-based electrodes. By examining the CV curves of Na_3_-MnO_2_ at varying potential ranges (Appendix A), it is clear that the CV curves at 0–1.0 V potential range and various scan rates from 2 to 50 mV s^−1^ have perfectly rectangular shapes at all scan rates with a small redox peak. Similarly, at 0–1.2 V, all the CV curves are rectangular, as shown in Appendix A. However, a prominent pair of redox peaks is observed in the CV curves at 0–1.3 V (Appendix A). The rectangular shape of enumerating the continuous and reversible faradaic redox transition with a pair of broad redox peaks at 0.4–0.6 V is noticeable in all the CV curves at all scan rates.

Figure 5c shows the comparison of typical CV curves of the Na_3_-MnO_2_ electrode at a scan rate of 10 mV s^−1^ in varying potential windows of 0–1.4, 0–1.3 0–1.2, 0–1.1, and 0–1.0 V, respectively. The CV curves of 0–1.0 and 1.1 V have an archetypal rectangular appearance with no redox peaks, which corresponds to the characteristic CV behavior for electrodes based on MnO_2_ [38,39]. The CV curve developed a very small redox peak by escalating the potential range to 1.2 V. By further increasing the upper cutoff potential to 1.3 V, we observed a prominent couple of redox peaks at around ≈0.4 V. Upon further extending the potential window to 0–1.4 V, two pairs of redox peaks emerged: one at the lower potential range at about 0.4–0.6 V and the other at the higher potential range of about 0.8–1.2 V. The first redox peaks are due to the characteristic charge storage mechanism of MnO_2_ by the intercalation and deintercalation of electrolytes on the Na_3_-MnO_2_ electrode [40,41]. However, the mechanism of the second redox peak differs from that of the first, as stated by Nawishta J. et al. [42] and Gang L. et al. [43]. The CV curve area of Na_3_-MnO_2_ at the 0–1.4 V potential window is much larger than that at 0–1.3 0–1.2, 0–1.1, and 0–1.0 V. A reversible redox reaction causes the current to rise at around 1.0 V [44].

As one can observe, irrespective of the potential range, the CV graph exhibited a pair of distinguishing redox peaks at about 0.8 V for the anodic scanning and 1.2 V for the cathodic scanning at a potential range of 0–1.4 V, which appeared due to fast reversible redox reaction of the Mn^3+^/Mn^4+^ plus the intercalation/deintercalation of the Na^+^. Remarkably, the potential windows sustain a fine rectangular shape even at a higher scan rate of 50 mV s^−1^ and 0–1.4 V. This justifies the fact that due to the insertion of the Na^+^, the charging potential window rises noticeably [35]. It offers the prospect of designing SCs with a much higher energy density [45].

Figure 5b demonstrates the GCD curves at a potential range of 0–1.4 V and varying current densities (5, 10, 15, 20, 30, 40, and 50 A g^−1^). The fact that all of these curves are completely triangular confirms the pseudocapacitive existence of the Na_3_-MnO_2_ electrode. The GCD curves at 0–1.0 V, 0–1.1 V, 0–1.2 V, and 0–1.3 V at current densities of 5, 10, 15, 20, 30, 40, 50, 80, and 100 A g^−1^ are shown in Appendix A, respectively. Figure 5d shows a comparison of GCD curves at 5 A g^−1^ current density at various potential ranges of 0–1.0, 0–1.1, 0–1.2, 0–1.3, and 0–1.4 V. Among these curves, the GCD curves at 0–1.4 V show a longer charge–discharge time. The charge–discharge time declined as the current density increased due to the reduced passage of electrolyte ions across the electrode [33]. Still, at a higher current density of 100 A g^−1^, the nanocomposite MnO_2_ electrode displayed a linear triangular shape, demonstrating that the pre-intercalation of Na^+^ leads to superlative electrochemical performance.

Figure 5e expresses a comparative analysis of specific capacitance of Na_3_-MnO_2_ at various potential windows (0–1.0, 0–1.1, 0–1.2, 0–1.3, and 0–1.4 V) as a function of current density. The specific capacitance of the Na_3_-MnO_2_ electrode at a potential window of 0–1.4 V can reach 546, 535, 490, 450, 390, 350, and 300 F g^−1^ at 5, 10, 15, 20, 30, 40, and 50 A g^−1^, respectively, which surpasses that of 480 F g^−1^ at the potential window of 0–1.3 V and is far better than 451 F g^−1^ at the potential window of 0–1.2 V at 5 A g^−1^ current density. The specific capacitance of the Na_3_-MnO_2_ electrode is much higher than the recently published work based on MnO_2_ electrodes, such as δ-MnO_2_ (251.4 F g^−1^ at current density 1 A g^−1^) [46], δ-MnO_2_ NFs@ α-MnO_2_ NWs (310 F g^−1^ at current density 1 A g^−1^) [47], MnO_2_@SBA-C (219 F g^−1^ at current density 1 A g^−1^) [48], 3D-HPCS@MnO_2_ (231.5 F g^−1^ at current density 1 A g^−1^) [49], δ-MnO_2_ (194.3 F g^−1^ at current density 1 A g^−1^) [50], NSs@MnO_2_ HNPAs (244.54 F g^−1^ at current density 1 A g^−1^) [51], α-MnO_2_@δ-MnO_2_ (206 F g^−1^ at current density 1 A g^−1^) [52], and rGO/CNT/MnO_2_ (209 F g^−1^ at current density 1 A g^−1^) [53]. At the same current density, the specific capacitance of the Na_3_-MnO_2_ electrode at the potential window of 0–1.1 and 0–1.0 V is 420 and 365 F g^−1^, respectively.

To comprehensively understand the stability of Na_3_-MnO_2_ electrodes in different potential ranges, the cycle stability of Na_3_-MnO_2_ is analyzed and correlated at a current density of 10 A g^−1^ over 6000 cycles and is shown in Appendix A. The Na_3_-MnO_2_ electrode has a capacitance retention of 95, 93, 94, and 96% in different working potential windows of 0–1.1, 0–1.2, 0–1.3, and 0–1.4, respectively. Noticeably, extending the potential window to 0–1.4 V has good cycling stability, which could be explained by the pre-intercalated Na^+^ in the Na_3_-MnO_2_ structure as a result of charge balancing, which acts as pillars to stabilize the nanosheets’ layered structures. The electrochemical performance of pure MnO_2_ was recorded to upgrade the understanding of the pre-intercalation of Na^+^ into MnO_2_ and its effect on the electrochemical behavior of the Na_3_-MnO_2_ electrode. Appendix A shows the CV of the MnO_2_ electrode at a scan rate of 1, 2, 5, 10, 20, 50, and 100 mV s^−1^ in different working potential windows of 0–1.0 (Appendix A), 0–1.2 (Appendix A), and 0–1.4 V (Appendix A), respectively. It is obvious that no redox peaks polarization occurred when the potential window exceeded 1 V. The current density of the CV curve was observed to be increased as the potential window was increased, without any observable redox peaks for all potential ranges. Furthermore, the CV curve of MnO_2_ at a potential range of 0–1.4 showed water decomposition with the evolution of oxygen. In addition, the area under the CV curve for Na_3_-MnO_2_ is larger than the current density of MnO_2_ at all potential windows. Similarly, the GCD analysis of MnO_2_ electrodes at different potential windows of 0–1.0 (Appendix A), 0–1.2 (Appendix A), and 0–1.4 V (Appendix A) at various current densities of 1, 2, 5, 10, 15, 20, 30, 40, 50, 80, and 100 A g^−1^ showed different results to Na_3_-MnO_2_. The most important point is that it did not even reach the potential window of 1.4 V. The GCD curves of Na_3_-MnO_2_ are more symmetric than those of MnO_2_, indicating the better reversibility of Na_3_-MnO_2_ electrode than that of the MnO_2_. Moreover, the discharge time of the Na_3_-MnO_2_ electrode is much longer than the pure MnO_2_ electrode, indicating higher capacitance of the Na_3_-MnO_2_ electrode. These results indicate that the pre-intercalation of Na^+^ can increase the capacitance performance and extend the stable potential window of MnO_2_ [54].

Similarly, Appendix A compares the specific capacitance of the MnO_2_ electrode as a function of current density at various potential ranges of 0–1.0, 0–1.2, and 0–1.4 V. MnO_2_ has a very low specific capacitance relative to the Na_3_-MnO_2_ electrode at all current densities. The highest recorded specific capacitance was noted to be 217 F g^−1^ at a current density of 1 A g^−1^ and a potential window of 1.4 V.

To test the applicability of the electrochemical performances of the Na_3_-MnO_2_ electrode in a full device for a wide potential window, an aqueous Na_3_-MnO_2_//N-AC device was engineered with a Na_3_-MnO_2_ electrode as the positive electrode (cathode) and commercial N-doped activated carbon (N-AC) electrode as the negative electrode (anode), completing this device with 1 M Na_2_SO_4_ solution as aqueous electrolyte. Before measuring the electrochemical performance of the Na_3_-MnO_2_//N-AC device, the electrochemical performance of N-AC was analyzed, as shown in Appendix A.

Appendix A depicts the CV curves for the Na_3_-MnO_2_ electrode and the N-AC in their respective potential windows of 1.4–0 V and 0–1.0 V at a scan rate of 10 mV s^−1^. Theoretically, it is clear from the separate potential window of cathode and anode that the Na_3_-MnO_2_//N-AC device could accomplish the working potential window of 0–2.4 V. To experimentally optimize the highest possible upper cutoff voltage for the Na_3_-MnO_2_//N-AC device, CV curves were reported at different potential windows from 1.6 to 2.6 V at a scan rate of 10 mV s^−1^, as shown in Appendix A. As the potential window of the Na_3_-MnO_2_//N-AC device extended beyond 0–2.4 V, a distinct current leap began to appear at a potential window of about 0–2.6 V, indicating the decomposition of the water started here. Notably, the CV curves of the Na_3_-MnO_2_//N-AC device could sustain a quasi-rectangular shape with potential windows ranging from 0 to 2.4 V, confirming the maximum potential window for the Na_3_-MnO_2_//N-AC device could reach as high as 2.4 V without any aqueous electrolyte decomposition.

To adjust the electrochemical performance of the Na_3_-MnO_2_//N-AC device, the mass ratio of the cathode to the anode was calculated to be 1.2:3.4, based on the charge balance theory (Equation (1)), at a current density of 1 A g^−1^. The electrochemical performance of the Na_3_-MnO_2_//N-AC device is shown in Figure 6.

CV curves of Na_3_-MnO_2_//N-AC device at various scan rates (5, 10, 20, 50, 100, and 200 mV s^−1^) in a wide (0–2.4 V) potential window were recorded and shown in Figure 6a. Interestingly, the quasi-rectangular shape of the CV curve was preserved at an even higher scan rate of 200 mV s^−1^, exhibiting the exceptional electrochemical characteristics of the Na_3_-MnO_2_//N-AC device. GCD measurements were used to further investigate the electrochemical properties of Na_3_-MnO_2_//N-AC device at 1, 2, 4, 8, 10, 15, 20, 30, and 50 A g^−1^ current densities and 0–2.4 V potential window (Figure 6b). All these GCD curves displayed quasi-triangular shape and good symmetry, confirming the ideal super capacitive behavior of the Na_3_-MnO_2_//N-AC device. From these GCD curves, the specific capacitances for the Na_3_-MnO_2_//N-AC device were calculated and are represented in Figure 6c. Fascinatingly, the Na_3_-MnO_2_//N-AC device has the highest specific capacitance of 80 F g^−1^ at a current density of 1 A g^−1^ and can retain 50% of its capacitance even at a high current density of 30 A g^−1^, indicating the excellent rate capability of the device.

According to Equations (1) and (2), the Na_3_-MnO_2_//N-AC device demonstrated the highest energy of 64 Wh kg^−1^ at the power density of 1225.53 W kg^−1^. This device maintained a higher energy density of 32 Wh kg^−1^ at a current density of 30 A g^−1^ even at the highest power density of 38400 W kg^−1^ (Figure 6d). These high-energy and power densities outperform those of recently published Mn-based devices, such as MnO_2_@PCN//PCN (energy density 31.13 Wh kg^−1^, at power density 193.6 W kg^−1^) [55], N/P-HCS@MnO_2_-30 (energy density 32.21 Wh kg^−1^, at power density 449.8 W kg^−1^) [43], G/MnO_2_ (energy density 19.6 Wh kg^−1^, at power density 351 W kg^−1^) [56], MnO_2_@SnO_2_//AC (energy density 18.05 Wh kg^−1^, 403.6 W kg^−1^) [57], MnO_2_@N-APC//N-APC (energy density 28 Wh kg^−1^, at power density 560 W kg^−1^) [58], AC//MnO_2_@NH_4_MnF_3_ (energy density 11.2 Wh kg^−1^, at power density 1000 W kg^−1^) [59], MnO_2_/SHAC-3//SHAC (energy density 46.2 Wh kg^−1^, at power density 98.5 W kg^−1^) [60], Fe-MnO_2_//AC (energy density 40 Wh kg^−1^, at power density 250 W kg^−1^) [61], and ov-MnO_2_@MnO_2_//AC (energy density 40.2 Wh kg^−1^, at power density 891.2 W kg^−1^) [62]. Furthermore, the quick and effortless procedure, low hydrothermal temperature, and shorter time spent formulating our samples make the materials and process strong and significant versus all the MnO_2_-based electrodes.

Appendix A shows the results of the volumetric energy and power density calculations. The Na_3_-MnO_2_//N-AC device delivered excellent volumetric energy and power density (12.8 Wh cm^−3^ at 245 W cm^−3^). Cycling performance is another very important parameter used to check the stability of the device. As noticed in Figure 6e, the cycling performance of the Na_3_-MnO_2_//N-AC device was recorded under a current density of 10 A g^−1^ for 10,000 cycles at a 2.4 V potential window. As seen in the graph, the device retained 83% of capacitance, confirming its outstanding cycling durability. Pre-intercalation could markedly mitigate these capacity fading concerns caused by structural instabilities during cycling because of the limited volumetric variability during the intercalation/de-intercalation phase of the carrier ions.

The EIS results of the Na_3_-MnO_2_//N-AC device are illustrated in Figure 6f, with a semicircle at a higher-frequency and a straight line at a lower-frequency region.

The electrochemical performance of Na ions pre-intercalated MnO_2_ was improved in this study. The most likely reasons are as follows. A broader interlayer space is particularly favorable for the intercalation–deintercalation of various ions during the redox process, which is given by wide interlayers of MnO_2_. The lattice space in the central region of the Na_3_-MnO_2_ nanosheets is 0.7 nm, which matches with the crystal plane (001) of δ-MnO_2_, as revealed by the HRTEM image, which supports the XRD results._._ The electrochemically active sites were increased due to the morphology with very thin nanosheets, which helped reduce the electron transport channel and the diffusion pathway of the ions of electrolyte, which optimized the electrochemical performance. Furthermore, the pre-intercalation of Na ions shrinks the nanosheets and disrupts the lattice alignment, as seen in the SEM images, resulting in a large turnout of grain borders. These numerous grain boundaries provide more active sites for redox reactions, as well as a simple diffusion pathway for ions and electrons, allowing the electrode materials to be used more competently. Furthermore, greater grain boundaries stimulate the formation of lattice defects, which increase the number of oxygen vacancies and so refine the electrical conductivity of Na_3_-MnO_2_.

## 4. Computational Studies

The present study was performed by implementing the density functional theory (DFT) using the generalized gradient approximation (GGA) and the Perdew–Burke–Ernzerhof (PBE)59 functional for the exchange correlation potentials that were accomplished through the Vienna ab initio simulation package (VASP)60-62. The structure was optimized using 8 × 6 × 4 k-points, and a 520 eV energy cutoff was applied. The electronic properties of bulk MnO_2_ were calculated using Dudarev’s PBE+U method [63], with U = 3.9 eV applied to the 3d states of each Mn atom. The U = 3.9 eV for Mn is also reported in the previous literature study [64,65]. Furthermore, the high symmetry and denser 16 × 12 × 10 k-points were used for the calculations of the bandstructure and density of states.

The MnO_2_ was modeled via a periodic cell containing 12 atoms (4 manganese and 8 oxygen atoms). The MnO_2_ supercell was orthorhombic 2.92 × 4.58 × 9.43 Å3 for the x, y, and z-directions, with α = β = γ = 90°, and the periodic condition was employed along the x, y, and z directions. The total energy was converged to an accuracy of 1 × 10^−5^ eV to obtain accurate forces, and a force tolerance of −0.02 eV/Å was applied in the structure optimization. The d orbital of manganese, p orbital of oxygen, and s orbital of Na hybridize, which results in the crowding of bands near the Fermi level in Na-intercalated hybrid structures c and d, exhibiting band gap reduction, which assists in charge transportation.

Figure 7c also exhibits the affinity between Na and oxygen of manganese oxide, therefore concluding that the intercalation of sodium is due to strong bonding between oxygen and sodium atoms.

To analyze the charge transfer and charge population, Figure 8, bader charge calculations were performed according to the following equation.
Work function = Vacuum potential (E_vac_) − Fermi Level (E_f_)(1)

The charge density difference (Δρ) was calculated as
Δρ = ρ δ − MnO_2_ − ρ Na
where ρ δ-MnO_2,_ ρ Na represents the charge densities of δ-MnO_2_ and Na-pre-intercalated δ-MnO_2._

The zero-point energy, entropy, and enthalpy corrections were added to the adsorbates to convert electronic energy to free energy. The Gibbs free energy (G) at 0 V is modified by Equation (2)
G = E_DFT_ + E_ZPE_ − TS + ∫ C_p_dT (2)
where E_DFT_, E_ZPE_, TS, and ∫ C_p_dT denote the electronic energy obtained from DFT optimization, zero-point vibrational energy, entropy, and heat capacity at room temperature.

The bader charge calculations prove that 7.40 |e| electrons are transferred from manganese to sodium, which may assist in the intercalation/de-intercalation process.

## 5. Conclusions

In conclusion, we designed Na^+^ pre-intercalated MnO_2_ for high-performance supercapacitors with a potential range of 0–1.4 V. The potential window of the Na_3_-MnO_2_ electrode can reach up to 1.4 V versus Hg/HgO. As a result, the Na_3_-MnO_2_ electrode has a high specific capacitance of 546 F g^−1^ and good rate capability. It demonstrated exceptional cycling stability with nearly 95% retention of the capacitance after 6000 cycles at 10 A g^−1^ at 1.4 V. Furthermore, the Na_3_-MnO_2_//N-AC device with Na_3_-MnO_2_ as positive and N-AC as negative electrodes demonstrated superior electrochemical performance with a wider potential window (0–2.4 V) and higher energy and power densities (64 Wh kg^−1^ at 1225.53 W kg^−1^), as well as excellent long-term cycling stability (83% capacitance retention after 10,000 cycles). This technique aims to bring new insights into the goal of a simple nanostructure, potentially leading to novel supercapacitor designs with longer working potential windows, more energy, and improved power density.

## Data Availability

The data are available on reasonable request from the corresponding author.

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
