# Peer review of "Sodium Pre-Intercalation-Based Na3-δ-MnO2@CC for High-Performance Aqueous Asymmetric Supercapacitor: Joint Experimental and DFT Study"

_nanomaterials, 2022, doi:10.3390/nano12162856_

Round 1

Reviewer 1 Report

This paper reports fabrication of Na3-δ-MnO2 on carbon cloth and their electrochemical performance as cathode materials for asymmetic supercapacitor. Although it is interesting, I can only  recommend it for publication after addressing these issues.

  1. Based on XRD analysis, the authors claim that "There is no discernible change in the position of diffraction peaks after the addition of Na ions". But when they interpreted the improved electrochemical performance of the Na3-δ-MnO2, they attributed this to the "broader interlayer space. I suggest the authors to clarify this by carefully comparing 2 theta of the (001) peak of MnO2 and Na3-MnO2.
  2. Based on Fig.2c, the authors state that the thickness of MnO2 nanosheets is a few hundreds of nm, while the thickness of Na3-MnO2 nanosheets is only a few nm. However, I could not distinguish this from the SEM image they provided. 
  3. What is the exact Na content in the Na3-MnO2 sample? I suggest them to measure Na content by ICP.
  4.  EIS test should be performed on MnO2 and Na3-MnOto confirm the boosted kinetics and better conductivity of Na3-MnO2.
  5.  In Fig.5, there are some duplicated figures. They should more carefully checking their manuscript before submission.
  6.  Based on the CV curves (Fig.S3a-d), a pair of broad redox peaks is always noticeable at 0.4-0.6 V. The authors should correct the corresponding description in the main text.
  7.  Some important and relevant papers should be cited properly, such as "Journal of Colloid and Interface Science, 2021, 588: 847-856", "Energy Fuels 2022, 36: 4596-4608" and "Journal of Materials Science & Technology, 2022, 99: 260-269".

Reviewer 2 Report

This article reported a one pot hydrothermal synthesis for sodium pre-intercalation MnO2cathode. The authors claimed that the cathode could deliver an extended potential window of 0-1.4V and specific capacitance of 546 Fg-1The authors also performed DFT studies on the MnO2 cathode and reported its electronic properties. However, there are some noticeable flaws (list below) about the DFT study part, And I do not recommend publishing this article on nanomaterials until a major revision is made.

Comment1: In page 12, the authors mentioned the equations about the calculations of work function (eq1) and Gibbs free energy (eq2). However, these two equations have no relation about the Bader charge calculation. And there is no relative discussion about these two equations in article.

Comment2: Also in the page 12, figure 7. (c). the Na intercalation energies of different sites should be compared to obtain the preferred intercalation site inside MnO2.

Comment3: In page 11, the valence electron number of different elements should be mentioned in the simulation details. 

Comment4: In page 7 and page 11, the figure 5 and figure 6 resolution should be increased.

Comment5: the interpretation about the band structure and density of state should be added.

Reviewer 3 Report

Sodium pre-intercalation based Na3-δ-MnO2@CC was successfully established via mild hydrothermal synthesis, and the as-prepared Na+ pre-intercalated δ-MnO2 nanosheets on carbon cloth exhibits a remarkable specific capacitance of 546 F g-1 at 5 A g-1, and its asymmetric aqueous supercapacitor illustrated an exceptional energy density (64 Wh kg-1 at 1225 W kg-1) and a good electrochemical cycle stability.  The construction strategy was facile and effective, the results were very good and interesting, and the manuscript was well prepared.    However, the manuscript still had some problems, therefore, it should be major revised before its acceptance. 

(1)  DFT Study is very important for this manuscript,  so “DFT calculation” should be used as one of the Keywords. 

(2)  I think that //soaked in deionized water for 10 minutes, and sonicated// should be “soaked and sonicated in deionized water for 10 minutes”. 

(3)  “Scheme 1 Synthesis of Na-MnO2@CC” was inconsistent with “2.2. Synthesis of Na3-MnO2” because the starting material was not δ-MnO2.  Modify it. 

(4)  The chemical reaction process/equation for the formation of Na3-MnO2 should be briefly clarified. 

(5)  The experimental evidence for confirming the stoichiometric ratio of Na3-MnO2 should be provided.

(6)  Line 126:  //uniformly grown on the carbon fabric (Fig. 2c)//  However, carbon cloth was used in the experiment for synthesis of Na3-MnO2 in the experiment.

(7)  Line 131-133:  //The presence of Na ions in MnO2 Nano sheets can be seen in the elemental mapping of the energy-dispersive X-ray spectroscope in Fig. S2.//    Na ions should be “sodium element”. 

(8)  All the graphs in Figure 4, Figure 5 and Figure 6 were severely deformed because the ratio of ordinate to abscissa is not 1 : 1.  Its readability was very poor. 

(9)  Line 192-193:  //Similarly, at 0-1.2 V, all the CV curves are rectangular, and no redox peaks are observed as shown in Fig. S3b.//    “no redox peak” should be changed into “no obvious redox peak”. 

(10)  The excellent works on high-efficient energy storage devices, as far as I know,  https://doi.org/10.1002/adma.202104148, https://doi.org/10.1016/j.jpowsour.2022.231512, https://doi.org/10.1039/D2TA00754A, are recommended to be referenced and also included in the References for balanced citations, providing more valuable information for the broader Readers. 

Round 2

Reviewer 1 Report

Although the authors made some modification, some of my concerns are not well addressed. I highly suggest them to consider my comments more carefully.

1. In Line 330-332, they claimed "A broader interlayer space is particularly favorable for the intercalation–deintercalation of various ions during the redox process, which is given by wide interlayers of MnO2 , as shown by XRD". Unfortunately, based on Fig.1, I cannot find any proof supporting their statement since there is no change in the 2 theta. How can you draw the abolve conclusion from the XRD spectrum? Beside, what do they mean by using the word "peak length"? 

2. The Fig2c and f do show some difference. But their statement "The thickness of Na0-MnO2 Nanosheets (Fig. 2c) is a few hundred nanometers, but the thickness of Na3 -MnO2 Nanosheets (Fig. 2f) is only a few nanometers"  was not right. If they insist on this, they must provide more clear SEM images which can support their statement. Otherwise, they should modify this discription.

3. To clarify the effect of Na pre-intercalation on difussion kinetics, the Nyquist plot of MnO2 and Na3-MnO2 should be compared.

Reviewer 3 Report

The manuscript had been well revised, and it can be accepted after minor revision.
Minor questions:
(1) //Fig. 1. XRD patterns of Na0-MnO2 and Na3-MnO2//.    “Na0” should be changed into “NaO”.  0 in “Na0” was zero.
(2) //Pure MnO2 was made using the same technique but without the addition of Na2SO4//    This sentence should be followed by a full stop.
Check the manuscript carefully to avoid any mistake.  
